# The Smokers Health Multiple ACtions (SMAC-1) Trial: Study Design and Results of the Baseline Round

**DOI:** 10.3390/cancers16020417

**Published:** 2024-01-18

**Authors:** Alberto Antonicelli, Piergiorgio Muriana, Giovanni Favaro, Giuseppe Mangiameli, Ezio Lanza, Manuel Profili, Fabrizio Bianchi, Emanuela Fina, Giuseppe Ferrante, Simone Ghislandi, Daniela Pistillo, Giovanna Finocchiaro, Gianluigi Condorelli, Rosalba Lembo, Pierluigi Novellis, Elisa Dieci, Simona De Santis, Giulia Veronesi

**Affiliations:** 1Faculty of Medicine and Surgery, School of Thoracic Surgery, Università Vita-Salute San Raffaele, 20132 Milan, Italy; antonicelli.alberto@hsr.it (A.A.); veronesi.giulia@hsr.it (G.V.); 2Department of Thoracic Surgery, IRCCS Ospedale San Raffaele, 20132 Milan, Italy; novellis.pierluigi@hsr.it (P.N.); dieci.elisa@hsr.it (E.D.); desantis.simona@hsr.it (S.D.S.); 3Department of Anesthesia and Intensive Care, IRCCS Istituto Oncologico Veneto (IOV), 35128 Padua, Italy; g.favaro@hotmail.it; 4Division of Thoracic Surgery, IRCCS Humanitas Research Hospital, 20089 Rozzano, Italy; giuseppe.mangiameli@cancercenter.humanitas.it (G.M.); fina.emanuela@hsr.it (E.F.); 5Department of Biomedical Sciences, Humanitas University, 20072 Pieve Emanuele, Italy; eziolanza@gmail.com (E.L.); giu.ferrante@hotmail.it (G.F.); gianluigi.condorelli@hunimed.eu (G.C.); 6Department of Interventional Radiology, IRCCS Humanitas Clinical and Research Center, 20089 Rozzano, Italy; manuel.profili@humanitas.it; 7Unit of Cancer Biomarkers, Fondazione IRCCS Casa Sollievo della Sofferenza, 71013 San Giovanni Rotondo, Italy; f.bianchi@operapadrepio.it; 8Cardio Center, IRCCS Humanitas Research Hospital, 20089 Rozzano, Italy; 9CERGAS and Department of Social and Political Sciences, Bocconi University, 20136 Milan, Italy; simone.ghislandi@unibocconi.it; 10Center for Biological Resources, Humanitas Cancer Center, IRCCS Humanitas Research Hospital, 20089 Rozzano, Italy; daniela.pistillo@cancercenter.humanitas.it; 11Department of Medical Oncology, Humanitas Cancer Center, IRCCS Humanitas Research Hospital, 20089 Rozzano, Italy; giovanna.finocchiaro@cancercenter.humanitas.it; 12Department of Anesthesia and Intensive Care, Section of Biostatistics, Università Vita-Salute San Raffaele, 20132 Milan, Italy; lembo.rosalba@hsr.it

**Keywords:** screening, early detection, lung cancer, smoking, tobacco, nicotine dependence, low-dose computed tomography scanning, coronary artery calcium, COPD, primary prevention

## Abstract

**Simple Summary:**

Lung cancer is still the leading cause of cancer-related death worldwide. The only recommended screening test for lung cancer is low-dose computed tomography (also called a low-dose CT scan). Early intervention is key as it improves the chances that treatments will lead to better prognoses. Various screening programs targeting high-risk individuals showed effectiveness in reducing the disease burden. Yet, in Italy, systematic ongoing prevention and control strategies are lacking, with no government at a national level. The SMAC Trial is an optimized multi-diseases screening program, with imaging, biological indicators, economic validation, general practitioner engagement, and awareness-to-patients elements. Thirty-two subjects were diagnosed with cancer, of which 30 were lung cancers (detection rate 2.7%); fourteen were cured (stage I on final pathology). We want our trial to stimulate further discussion among policy-makers to implement our efforts. In fact, lung cancer screening extends far beyond an imaging: a well-organized and comprehensive program is vital to ensuring high-quality and timely care through screening, diagnosis, and treatment. Thus, financial incentives for lung cancer screening sites are needed, too. As new technologies continue to emerge aiming to change lung cancer patients’ diagnostic and treatment journey, even more so early-detection via screening requires a continued support nationally.

**Abstract:**

Background: Lung cancer screening with low-dose helical computed tomography (LDCT) reduces mortality in high-risk subjects. Cigarette smoking is linked to up to 90% of lung cancer deaths. Even more so, it is a key risk factor for many other cancers and cardiovascular and pulmonary diseases. The Smokers health Multiple ACtions (SMAC-1) trial aimed to demonstrate the feasibility and effectiveness of an integrated program based on the early detection of smoking-related thoraco-cardiovascular diseases in high-risk subjects, combined with primary prevention. A new multi-component screening design was utilized to strengthen the framework on conventional lung cancer screening programs. We report here the study design and the results from our baseline round, focusing on oncological findings. Methods: High-risk subjects were defined as being >55 years of age and active smokers or formers who had quit within 15 years (>30 pack/y). A PLCO_m2012_ threshold >2% was chosen. Subject outreach was streamlined through media campaign and general practitioners’ engagement. Eligible subjects, upon written informed consent, underwent a psychology consultation, blood sample collection, self-evaluation questionnaire, spirometry, and LDCT scan. Blood samples were analyzed for pentraxin-3 protein levels, interleukins, microRNA, and circulating tumor cells. Cardiovascular risk assessment and coronary artery calcium (CAC) scoring were performed. Direct and indirect costs were analyzed focusing on the incremental cost-effectiveness ratio per quality-adjusted life years gained in different scenarios. Personalized screening time-intervals were determined using the “Maisonneuve risk re-calculation model”, and a threshold <0.6% was chosen for the biennial round. Results: In total, 3228 subjects were willing to be enrolled. Out of 1654 eligible subjects, 1112 participated. The mean age was 64 years (M/F 62/38%), with a mean PLCO_m2012_ of 5.6%. Former and active smokers represented 23% and 77% of the subjects, respectively. At least one nodule was identified in 348 subjects. LDCTs showed no clinically significant findings in 762 subjects (69%); thus, they were referred for annual/biennial LDCTs based on the Maisonneuve risk (mean value = 0.44%). Lung nodule active surveillance was indicated for 122 subjects (11%). Forty-four subjects with baseline suspicious nodules underwent a PET-FDG and twenty-seven a CT-guided lung biopsy. Finally, a total of 32 cancers were diagnosed, of which 30 were lung cancers (2.7%) and 2 were extrapulmonary cancers (malignant pleural mesothelioma and thymoma). Finally, 25 subjects underwent lung surgery (2.25%). Importantly, there were zero false positives and two false negatives with CT-guided biopsy, of which the patients were operated on with no stage shift. The final pathology included lung adenocarcinomas (69%), squamous cell carcinomas (10%), and others (21%). Pathological staging showed 14 stage I (47%) and 16 stage II-IV (53%) cancers. Conclusions: LDCTs continue to confirm their efficacy in safely detecting early-stage lung cancer in high-risk subjects, with a negligible risk of false-positive results. Re-calculating the risk of developing lung cancer after baseline LDCTs with the Maisonneuve model allows us to optimize time intervals to subsequent screening. The Smokers health Multiple ACtions (SMAC-1) trial offers solid support for policy assessments by policymakers. We trust that this will help in developing guidelines for the large-scale implementation of lung cancer screening, paving the way for better outcomes for lung cancer patients.

## 1. Introduction

Despite some incremental advances, lung cancer is still the world’s deadliest malignancy [1,2]. It can be indeed intrinsically aggressive and, tragically, even to this day there is a lack of systematic early detection [3]. Enabling care at the earliest possible stage is a crucial public health strategy [4,5]. Every possible effort must be made to avoid a shift in stage, as even going from T1a to T1b means a significant change in prognosis. Unfortunately, most patients are asymptomatic; hence, early diagnosis cannot rely on patients’ awareness of early symptoms. This is why there is a need for tools capable of ensuring early lung cancer detection, paving the way to potentially curative treatments [6,7].

Low-dose computed tomography (LDCT) is gaining constant traction as a screening tool for lung cancer [8]. International scientific societies have indeed developed protocols for high-risk subgroups (active or former smokers aged 50 years or older) [9,10,11,12,13,14,15]. In the USA, the Centers for Medicare & Medicaid services (CMS) has recently announced a national coverage determination that expanded reimbursement for lung cancer LDCT screening to improve health outcomes for people with lung cancer [15]. In China, even without reimbursement, most people can afford the expense since LDCT is both cheap (~ USD 30) and easy to access.

Patients who are eligible for LDCT lung cancer screening are also at high-risk of cardiovascular diseases (CVD) [16]. LDCT allows for the assessment of coronary artery calcification (CAC), which may help further stratify cardiovascular risk. Furthermore, CAC can be assessed within the same LDCT screening examination [17,18]. Lung cancer is also intimately related to chronic obstructive pulmonary disease (COPD), with great impact on public health [19]. There is a two- to four-fold increase in lung cancer risk in asymptomatic subjects with (and even without) a prior diagnosis of COPD. The simultaneous evaluation of CAC and emphysema in LDCT scans offers an unprecedented opportunity to include CVD and respiratory risk assessments in lung cancer screening programs [20,21,22,23,24,25]. Nevertheless, limited data exist about the value of implementing a cardiovascular and respiratory primary prevention screening program among those subjects who are undergoing LDCT for lung cancer screening already.

Cost-effectiveness analyses (CEAs) for lung cancer screening have produced conflicting results [26,27,28]. There are no CEAs for programs combining lung cancer screening with smoking cessation counselling, CVD prevention, and early treatment of COPD [29,30]. Furthermore, early-detection screening by LDCT for lung cancer, CVD, and COPD could benefit from the addition of clinical diagnostic markers, such as those present in the blood circulation [31,32,33,34,35,36,37,38].

The overarching goal of our project, the Smokers health Multiple ACtions (SMAC-1) trial, was two-fold: (i) to show how an integrated lung cancer, CVD, and COPD prevention program, alongside primary prevention, can increase early detection of disease states, thus enabling early treatment of high-risk patients and optimizing health-care costs; (ii) to support Italian policy makers in providing our national healthcare system (NHS) with screening programs for such highly fatal diseases.

Here, we report the study design and the results from the baseline round of our new multicomponent LDCT-based lung cancer screening program in 1112 consecutive high-risk subjects, focusing on oncological findings.

## 2. Methods

Subjects’ accrual occurred prospectively for one year (September 2018 through September 2019). The general population was asked to send by email (smac@humanitas.it) their willingness to participate to the project.

Subjects needed to be at high-risk for lung cancer; therefore, the following inclusion criteria were set: >55 years of age, active smokers or formers who had quit within 15 years (>30 pack/y), PLCO_m2012_ threshold risk >2% in 6 years [39]. Exclusion criteria were having received LDCT within the last 18 months, previous participation in other lung cancer screening programs, severe lung or extra-pulmonary disease. The SMAC-1 project study involved human participants and was approved by the Human Research Ethics committee of the IRCCS Humanitas Research Hospital in March 2018 (ethics approval number: 2123; ClinicalTrials.gov Identifier: NCT04315766). All subjects signed an informed consent prior to the enrollment. SMAC-1 encompassed three objectives: (1) to build the infrastructure of a screening program combining measures to prevent lung cancer, CVD, and COPD; (2) to validate circulating biomarkers to enhance the sensitivity and specificity of LDCT for lung cancer, CVD, and COPD; (3) to perform a cost-effectiveness analysis (CEA) of the screening program.

For the recruitment phase to be considered successful, our goal was to get at least 30% of the eligible subjects to join the screening. To implement accrual, there was a collaborative effort to involve General Practitioners (GPs) from our local health authority, namely the ATS (Agenzia di Tutela della Salute) of the Metropolitan city of Milan and its surrounding districts. Informative material and accrual modalities were sent by e-mail to more than 2000 GPs. Furthermore, dedicated conference meetings were organized to raise awareness of primary and secondary lung cancer prevention, obtaining high attendance by GPs and exposure by the media. Also, public campaigns at regional and national levels were organized to allow subjects of any social class to attend, with the IRCCS Humanitas Research Hospital being the major hosting and coordinating institution [40]. The project infrastructure is shown in Figure 1.

Eligible subjects were contacted through a dedicated call center and a full-day visit was scheduled, including a thoracic surgery consultation, to explain the details of the project and obtain written consent; a research nurse consultation, to obtain a blood sample, a questionnaire [39]; a spirometry and peripheral oxygen saturation (SpO_2_) monitoring by pulse oximetry; a psychology consultation of 20 min in order to create a space both for the subject (to feel at ease to share his/her relationship with tobacco use) and for the psychologist (to identify the main triggers, the type of physical dependency—Fagerström test, and the subject’s motivation to quit smoking—Motivation to Quit test derived by Test di Marino [41]); to measure the exhaled CO (piCO Smokerlyser, Bedfont Scientific, Harrietsham, UK); and to encourage attendance of anti-smoking clinics. Along with psychological counselling, a pharmacological approach using cytisine was offered for 46 days to help control symptoms from nicotine withdrawal [42].

To conclude the full-day visit, baseline LDCT scanning was performed. The CT scanner was a Brilliance 64 multi-detector row (Philips, Hamburg, Germany; 32 or more slices, 100 kV, 50 mAs, 1 mm slice thickness). Two dedicated radiologists read the LDCT images. Subjects found to have at least one non-calcified lung nodule greater than 6 mm (max diameter, i.e., including the non-solid component, if any) were discussed at the MDT meeting. For each nodule, the size, volume, consistency, presence of calcifications, as well as other incidental findings in the chest, were recorded. Then, the dedicated diagnostic algorithm was applied (Figure 2). Nodules that looked to be suspicious enough to warrant further testing as early as possible, were evaluated for preoperative diagnosis, followed by surgical resection if indicated. Multimodality treatment was indicated for advanced stages of disease [43,44]. Patients thus received guideline recommended oncological follow up [44]. Subjects without suspicious findings were scheduled for annual CT scans for two additional years or for biennial CT scan, according to the Maisonneuve risk score [45].

CAC score: A cardio-imaging radiologist performed coronary artery calcium quantification with non-gated LDCT scan on all subjects screened for lung cancer. Coronary artery calcium (CAC) was quantified using the Agatston score (a sum of the attenuation in Hounsfield units), and five groups of no (0), mild (1–99), moderate (100–299), high (300–999), and very-high (≥1000) CAC score were defined.

Cardiovascular risk: The 10-year risk of fatal cardiovascular events was assessed by the HeartScore, recommended by the European Society of Cardiology at the time of the study.

### 2.1. Translational Research Associated to Our LDCT Screening Program

Circulating biomarkers: For each subject, 10 mL of whole blood was collected and an average of 7 aliquots of 0.3 mL of plasma were then prepared to analyze circulating biomarkers, as follows: (a) Circulating tumor cells (CTCs) as early detectors of lung cancer, alongside the relationship between their number and both disease stage and prognosis; microRNAs (miRNAs) expression in regulating pentraxin-3 (PTX3) protein levels as early detectors of lung cancer and COPD. Specifically, a ‘Taqman OpenArray Human Advance Microrna Custom panel’ (ThermoFisher, Waltham, MA, USA) was designed, composed of 10 CHIP to allow the analysis of 56 cf-miRNAs (including the 45 cf-miRNAs signature) A synthetic miRNA (the Arabidopsis thaliana miR159a) was also added before extraction to each sample as a spike-in to control for plasma miRNA extraction efficiency. (b) Circulating miRNAs for lung cancer early detection [38,46,47,48,49]. (c) miRNAs as prognostic biomarkers of major cardiovascular events. (d) Interleukins, i.e., IL-2, IL-5, IL-8, and IL-13 as early detectors and prognostic factors of COPD. Further blood samples were collected from those subjects with benign nodules. For cancer patients who underwent tumor resection surgery, an additional 28 mL whole blood was collected before surgery and, when possible, tumor and non-tumor tissue was obtained as excess material from material resected during surgery and not used for histological diagnosis. The tissues obtained were immediately frozen in liquid nitrogen and stored at −80 °C, according to the procedures of the Humanitas Biological Resources Center, for later analysis. The results of these studies will be discussed in manuscripts that are currently under preparation.

Cost-effectiveness analysis: To evaluate the direct costs for the detection, diagnosis, and treatment of smoking-related diseases, clinical data from both screened and not screened (controls) patients were collected. Associated direct costs from the NHS perspective were derived using the Italian national tariffs and public employment salaries [50,51,52].

Furthermore, the model simulates the costs associated with different invitation strategies, from simple letters of invitation to active GP involvement with targeted phone calls.

The incremental cost-effectiveness ratio (ICER) per quality-adjusted life years (QALYs) gained was calculated using a dynamic cohort-based Markov model with two components: (1) the natural history of disease progression and (2) the treatment and aftercare pathways based upon the lung cancer stage at diagnosis. The ICER and net monetary benefit (NMB) of four specific LDCT-based screening invitation scenarios were compared to standard clinical care. Costs and QALYs were used as outcomes to assess a health opportunity cost threshold. Deterministic and probabilistic sensitivity analyses were conducted.

A database was built using w-Hospital (v. 28.4.5.1289; 2023), a data repository software provided by IRCCS Humanitas Research Hospital. Multiple rounds of data cleaning and requests for missing data were performed to obtain a database that was as complete as possible. Deidentified data were extracted on 30 April 2020 and the final dataset was used for analysis.

### 2.2. Statistical Analysis

Continuous variables were described using mean, standard deviation, median, and interquartile values. For discrete variables, relative and percentage frequency were reported.

Pearson’s correlation was used to analyze the relationship between the selected variables to measure the strength and the direction of such relationships.

## 3. Results

### Overview

Subject selection: A total of 3228 subjects answered our screening call. Of these, only 194 (6%) were sent by their GPs and 1654 were scored as high-risk subjects and thus were eligible for the screening program. A total of 1112 subjects finally participated in the SMAC-1 study and their demographic characteristics are reported in Table 1.

Management of identified nodules: At least one nodule was identified in 348 subjects (31.2%) (Table 1). The maximum and average number of nodules per patient was 10 and 1.56, respectively, for a total of 539 lung nodules. Within these, 469 nodules showed a solid/partially solid component (373 and 96, respectively), while 62 showed a non-solid component (Figure 3).

A total of 107 subjects presented more than one nodule (9.6%); in such case, data analysis was performed considering the most clinically relevant one. A total of 217 nodules were found in the right lung (98 in the upper, 87 in the lower, and 37 in the middle lobes: 9%, 8%, and 3.3%, respectively), while a total of 122 nodules were found in the left lung (76 in the lower and 46 in the upper lobes: 7% and 4%, respectively).

Nodules were further subdivided considering their maximum diameter (including the non-solid component, if any [53]), as follows: Ø < 6 mm, Ø 6–8 mm, or Ø > 8 mm for those with a solid/partially solid component (Table 2, a), and Ø < 6 mm, Ø 6–14 mm, or Ø > 14 mm for those with a non-solid component (Table 2, b).

Baseline LDCTs showed no clinically significant findings in 762 subjects (69%), of which 143 had a Maisonneuve risk score ≥ 0.6%; hence, they were referred to annual LDCTs. Contrariwise, 619 subjects had a Maisonneuve risk score < 0.6%; hence, they were referred to biennial LDCTs. The mean Maisonneuve risk score was 0.44% (±0.22% SD).

The recall rate was 13.4% (149 subjects). Lung nodule surveillance was indeed indicated for 122 subjects (11%): 13 at one month, 93 at three months, and 16 at six months. During this period, 15 subjects were referred to further LDCT scans (“early” surveillance, also called active surveillance), 15 were lost to follow up, 44 underwent a PET-FDG, and 27 underwent a CT-guided transthoracic fine needle aspiration biopsy (FNAB) (Figure 3).

The final number of subjects diagnosed with cancer was 32, of which 30 were lung cancers (i.e., detection rate 2.7%) and 2 were extrapulmonary cancers (i.e., malignant epithelioid pleural mesothelioma and thymoma). Twenty-five patients underwent lung surgery, out of which fourteen were diagnosed with stage I disease in the final pathology report (Table 3).

Two patients received neoadjuvant chemotherapy for stage IIIA disease (cT2N2M0 and cT1bN2, re-staged as ypT2bN0 and ypT1bN2, respectively), while four underwent adjuvant treatment (one chemotherapy, three chemo-radiotherapy). Importantly, there were no false positives, while two false negatives occurred with CT-guided biopsy, of which the patients were operated on with no stage shift. Five patients did not undergo surgical treatment, and three of these with had an indication for exclusive concurrent radiochemotherapy, one for systemic chemotherapy alone, and one for palliative care.

The median surveillance was 15 months (IQR 13–17). Due to the COVID-19 pandemic, continued follow up was not feasible firstly for those subjects with a baseline LDCT scan clinically not significant and secondly for patients with nodules considered to be low risk. On the other hand, more than 90% of subjects with nodules considered suspicious were re-called and did re-present, showing themselves to be compliant with MDT instructions.

As the risk for developing lung cancer calculated with the PLCO_m2012_ risk model increased, the same risk calculated with the Maisonneuve model also increased (Pearson’s correlation coefficient = 0.31, *p* < 0.0001). The risk of lung cancer calculated at baseline (PLCO_m2012_ risk model) and at subsequent screening (time interval determined by the Maisonneuve model) correlated significantly with age (0.55 and 0.17, respectively, both *p* < 0.0001).

Finally, the results of primary prevention, CVD and COPD assessment, circulating biomarkers analysis, and CEA are under evaluation and will be reported and commented on in dedicated papers.

## 4. Discussion

Tobacco use is the single greatest preventable cause of disease and premature death worldwide. At the time of diagnosis, lung cancer is often already at an advanced stage, with a 5-year survival of 15% or less [54,55,56,57].

Early detection is a powerful cancer-fighting weapon with two core advantages: (i) early-stage lung cancer can be cured, and (ii) in most cases it can be treated with minimally invasive surgery (MIS), sparing surrounding healthy tissue with an enhanced post-operative recovery [7]. And yet, the lung cancer screening rate has increased by only 2% over the past decade in the U.S. [58], from 4% in January 2010 to 6% in January 2021. Possible reasons include onerous eligibility requirements like “shared decision making”, which no other cancer screening test has to meet.

Despite having several screening programs in our community, we still see screen-eligible patients present with symptomatic advanced lung cancer having never undergone lung cancer screening. This is particularly disappointing when for years patients have been diligently getting mammograms, colonoscopies, Pap smears, and serial ultrasounds for stable thyroid nodules. As of today, LDCT lung cancer screening seems to be the “redheaded stepchild” of screening exams.

Successful intervention begins with identifying users and appropriate interventions based upon the patient’s willingness to quit. This is also why the Smokers health Multiple Actions (SMAC-1) study aimed at a transversal dissemination action. The already well-known risks for lung cancer and other smoking-related diseases were taken just as the starting point to make LDCT screening visible to a wider community of doctors, namely GPs (family doctors), cardiologists, and pneumonologists. SMAC-1 intended to educate on and sensitize GPs to the role of primary and secondary lung cancer prevention with LDCT screening. For example, we strived to train GPs through dedicated oral sessions and webinars on their potential key role in fighting lung cancer to ultimately implement smoking cessation activities, e.g., through the application of the “5 A’s” [59]. We believe indeed that family medicine practitioners must make treating tobacco dependence a top priority [51].

There are three types of recruitments for screening: active, where a high-risk subject is identified with the goal to set a first appointment; voluntary, where the subject reaches out to the screening program autonomously; and GP based. An implementation to active recruitment can be found in the Targeted Lung Health Check (TLHC) program, where effective communication reduced the risk of community stigmatization and mobile scanners were used close to social facilities [60,61,62,63]. Regarding the SMAC-1 recruitment phase, 67% of eligible subjects joined the screening. This was mainly through media and social media campaigns. Unexpectedly, GP-based recruitment was very low, as only a 6% response rate was obtained (194 subjects). As we all become busier and busier navigating administrative burdens in the paperwork crisis, perhaps little time is left for GPs to include lung cancer screening engagement within a broader assessment of prioritized patient needs [64,65]. Although GPs remain a fundamental bridge to patients at risk for any cancer, as of today, in Italy, patients at risk for highly fatal cancers like lung cancer cannot rely on a GP-based recruitment alone. Alternative methods are warranted. This sheds light on the importance of an institutional-based recruitment, i.e., the Italian NHS (SSN), where institutions are directly involved in reaching out to high-risk individuals. A ‘population mail-out’ strategy could, for example, increase screening participation dramatically as it is a convenient, cost-effective, and sensitive method [66]. Importantly, our diagnostic algorithm (Figure 2) led to no false positives at surgery. Those subjects who displayed a negative LDCT at baseline or small indeterminate nodules (<6 mm) were referred to receive a 12- or 24-month LDCT based on the Maisonneuve risk calculation, in contrast to a fixed 12-month timepoint (Lung-RADS categories 1 and 2) [67]. Those subjects who showed suspicious nodules by size (>6 mm), volume, or morphology, e.g., spiculation (Lung-RADS categories 3–4), were discussed at our MDT meeting, wherein the time-interval for early surveillance (LDCTs at 1, 3, or 6 months) and/or the need for second-level diagnostics like PET-FDG and/or FNAB were decided. Furthermore, volumetric measurements, namely Volume Doubling Time (VDT) [68], were another important factor guiding us in the diagnostic workup. There were two false negatives with the CT-guided biopsy, and yet they were highly suspicious with the PET-18 FDG. Thus, surgical resection was performed anyway. This may have led to a slight diagnostic delay, but the final pathology report stated that they both were T1N0 lung cancers. Stage I detection, avoiding a shift in stage, is indeed the goal of a lung cancer screening program.

Notably, the surgical approach for screening-detected lung cancers has changed over time [69] (Figure 4). On the one hand, the National Lung Screening Trial (NLST), the first of its kind, showed that 61% of the surgeries were performed by thoracotomy (missing data, 9%). On the other hand, the Danish Trial [9,10,11] showed that MIS was preferred over the open approach (84% and 16%, respectively). MIS results in less post-operative pain via a reduction in the immune-mediated inflammatory reaction [70,71,72], resulting in early discharge; quicker functional recovery, and hence rapid return to daily living activities [73,74,75]; and improved aesthetic results [76]. In our study, surgeons preferred an MIS approach, either by RATS or VATS, to thoracotomy (84 vs. 12%), as in the Danish trial, but with RATS being the most predominant (56%). Despite the relatively recent introduction of the surgical robotic system, it has gained substantial traction over the last 10 years. More surgical robotic systems are becoming commercially available [77]. We see RATS indeed as the evolution of manual VATS, bearing widely described technical advantages such as a better definition of the operating field (3D vs. 2D).

Furthermore, we compared the extent of lung resection among the NLST/Danish trials [5,7] and our trial (Figure 4). In the formers, the number of segmentectomies was very low (2 and 5%, respectively); in the latter, seven segmentectomies (28%), one wedge resection (4%), and seventeen lobectomies (68%) were performed. In modern screening studies, such as SMAC-1, patients are indeed operated on using lung-sparing techniques, i.e., radical segmentectomy. Sublobar resections (SLR) are destined to rapidly increase thanks to the results of two studies: the JCOG 0802/WJOG 4607L [78] and the CALGB/Alliance 140503 [79], which showed similar outcomes between SLR and lobectomy for NSCLC ≤ 2 cm. In the former, segmentectomy showed better overall survival, due to a reduced mortality for other causes, despite an increased rate of local recurrence.

Lung cancer is a disease that mostly remains asymptomatic at length; hence, it is often diagnosed at an advanced stage. So, we compared the results on pathological staging from the present study with another trial: the Continuous Observation of Smoking Subject (COSMOS 1), also carried out in Milan, Italy, fifteen years before ours [4,29,80,81] (Figure 5).

In the COSMOS study, 5203 high-risk subjects (age ≥ 50, smoking pack/Year ≥ 20, abstinence years ≤ 10) underwent baseline LDCT and subsequent annual rounds of LDCT. Lung nodule management included the evaluation of VDT and the use of PET-FDG. At the one-year surveillance point, 55 lung cancers were detected (detection rate 1.1% vs. 2.7% for SMAC-1). Then, at five, six- and ten-years surveillance, 175, 196, and 259 lung cancers were detected, respectively (detection rates 3.4%, 3.8%, and 5%, respectively). After one year, fewer stage I diseases (47 vs. 65%), in favor of more advanced stages (53 vs. 19%), were reported in the former (*p* = 0.059). We believe that selecting a population with a higher risk (PLCO_m2012_ threshold > 2%), as selected in SMAC-1, led to an increase in the number of more aggressive tumors according to the correlation between the VDT of tumors and the individual risk based on the Maisonneuve model [68].

For screening selection, a PLCO_m2012_ threshold risk > 2% was used indeed. The decision of a slightly higher threshold than the ones used in other previous studies was justified by our goal to detect a higher number of cancers in a shorter period. The average risk of each patient is related to the detected stage. In other words, with a higher threshold, the detected stage will be higher as well. SMAC-1 provides the first validation of this threshold in Italy. Re-calculating the lung cancer risk with the Maisonneuve model allowed us to optimize time intervals to subsequent screening. Most subjects (81%) with no clinically significant findings at the baseline LDCT scan were indeed referred to biennial LDCT scanning, confirming the results from the MILD trial [82] and thus reducing all screening-induced harms, i.e., radiation exposure, psychological stress, and costs both for the participant and for the hospital.

Translational relevance and impact for the Italian National Health System (SSN): International screening programs have already documented a positive role of LDCT screening in reducing lung cancer-specific mortality. On a European level, there is a clear consensus on the need for lung cancer screening implementation [83]. SMAC-1 represents an evolution of the state of the art as it evolved into a multi-disease screening program empowered by predictive bio-marker assessments (separate manuscript under preparation), and it was intended to be combined with patient education thanks to family physicians’ engagement. Collaboration is key in building strong relationships, which allows organizations, teams, and individuals to support each other. This is why we believe that despite GP-based recruitment not having much success, our project still sets a strategic example for the implementation and application of lung cancer screening at a regional (and even national) level.

A personalized time-interval of screening according to the Maisonneuve model was also implemented in this study. The model described in 2011 [12] was validated in a second cohort of screening subjects [45] and prospectively implemented here. The results of the follow up will be described in a separate paper.

The simultaneous evaluation of CAC and emphysema in LDCT scans offers an unprecedented opportunity to include CVD and respiratory risk assessments in lung cancer screening programs [20,21,22,23,24,25], in line with new guidelines from the European respiratory society on collateral findings [83,84,85,86,87].

By combining clinical risk variables with a gene-based risk score, even greater reductions in lung cancer mortality can be achieved with LDCT scans [88,89,90]. Biomarker-led outcome-based approaches may help to better define which eligible smokers might defer screening (low risk of lung cancer), discontinue screening (high risk of overtreatment with little benefit), or continue screening to achieve the greatest reduction in lung cancer mortality. The development of a blood test based on serum/plasma biomarkers, e.g., circulating miRNAs, will indeed make screening easier and safer to participate in, at reduced costs, due to the absence of radiation exposure. The same concept applies to diagnosis as molecular fingerprinting of blood-based biopsies may avoid attempts to diagnose the T directly, e.g., via CT-guided FNAB, which is invasive by definition and, therefore, may even require hospitalization.

Finally, SMAC-1 proved LDCT screening to be cost effective in Italy [91]. The ICER per QALY gained for screening-detected lung cancer is lower than that for advanced-stage treatments [92,93,94,95,96]. Also, in Italy, the yearly cost for disease prevention is EUR 5 billion. Less than 0.5% of this sum would be enough to screen 400,000 individuals per year. This use of resources becomes even more important when we consider that lung cancer risk decreases by 39% five years after quitting smoking, but after 25 years is still three times the risk of never smokers [97]. SMAC-1 represents the first sample-based economic assessment of a targeted multi-disease screening program in Europe. Our goal was indeed to stimulate the debate on current policies and to improve existing screening [90]. We hope that our pilot study can be validated and further implemented to finally demonstrate that upfront investments in lung cancer screening at a national level can also optimize those resources needed to manage other smoke-related contributions to mortality, such as CV and respiratory diseases.

## 5. Conclusions

Low-dose computed tomography continues to confirm its efficacy in safely detecting early-stage lung cancer in high-risk subjects, with a negligible risk of false positives. In the Smokers health Multiple Actions (SMAC-1) trial, no patient underwent surgical intervention for benign disease (false positives at surgery = 0%). Re-calculating the lung cancer risk with the Maisonneuve risk model allows one to optimize time intervals to subsequent screening in subjects with no clinically significant findings at baseline LDCT scanning [45].

LDCT screening can lead to a reduction in all-cause mortality (including lung cancer related). The SMAC-1 trial, in fact, includes an optimized multi-diseases screening, with imaging and biological indicators, economic validation, GP engagement, and awareness-to-patients elements. A lung cancer (LC) screening program incorporating smoking cessation, cardiovascular prevention, and early treatment of COPD can dramatically reduce mortality and morbidity, with these three elements being the three main causes of death and disability. The inclusion of innovative molecular and cytological biomarkers could further improve the sensitivity and specificity of LC screening programs, providing unprecedented benefits to high-risk subjects and to the economy. Finally, SMAC-1 can offer solid support for policy assessments by policymakers, payers, and guideline developers who are faced with the important decision of whether to implement population-based lifesaving lung cancer screening programs.

Detection starts with screening, and screening starts with education. We trust that this altogether will help to develop guidelines for the large-scale implementation of lung cancer screening, paving the way for better outcomes for lung cancer patients.

## Figures and Tables

**Figure 1 cancers-16-00417-f001:**
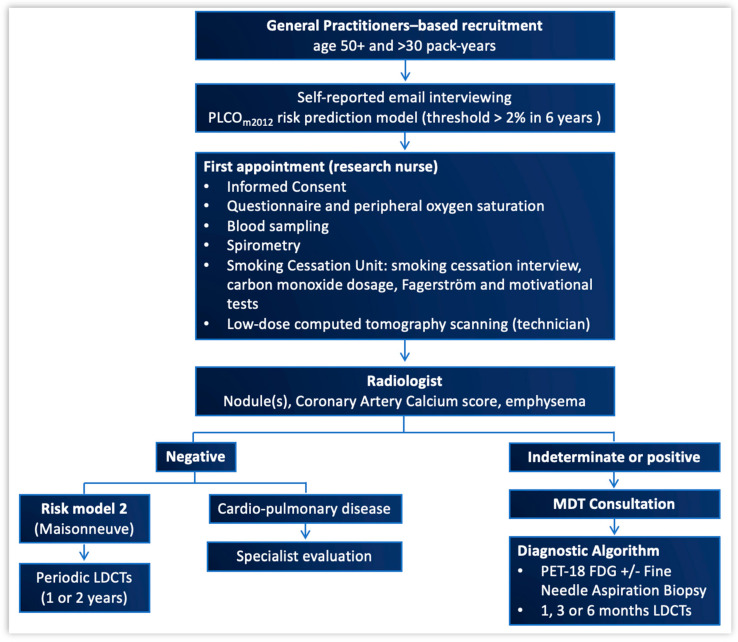
Subject’s journey: from recruitment, through the first appointment, to possible lung nodule management. MDT: multidisciplinary team (including the following elements: thoracic surgeon, radiologist, anatomopathologist, oncologist, radiotherapist, pneumonologist, cardiologist, research nurse, psychologist, radiology and biobank technicians, data manager).

**Figure 2 cancers-16-00417-f002:**
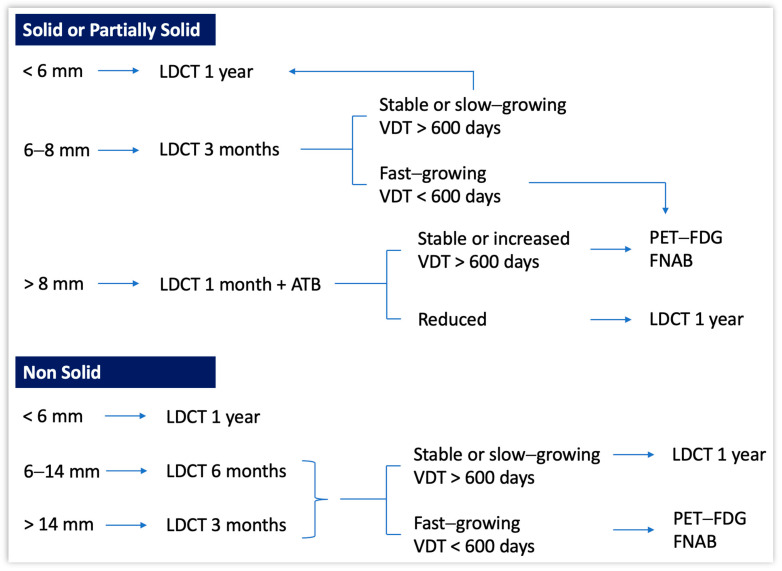
Diagnostic algorithm for non-calcified nodules by morphology, integrating size, and Volume Doubling Time (VDT) at baseline LDCT. ATB: antibiotic therapy.

**Figure 3 cancers-16-00417-f003:**
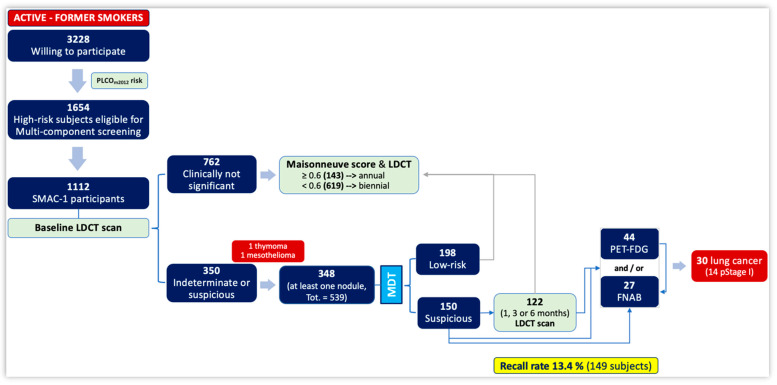
Subjects’ enrollment, lung nodules detection and management, follow up, and final diagnoses in SMAC-1.

**Figure 4 cancers-16-00417-f004:**
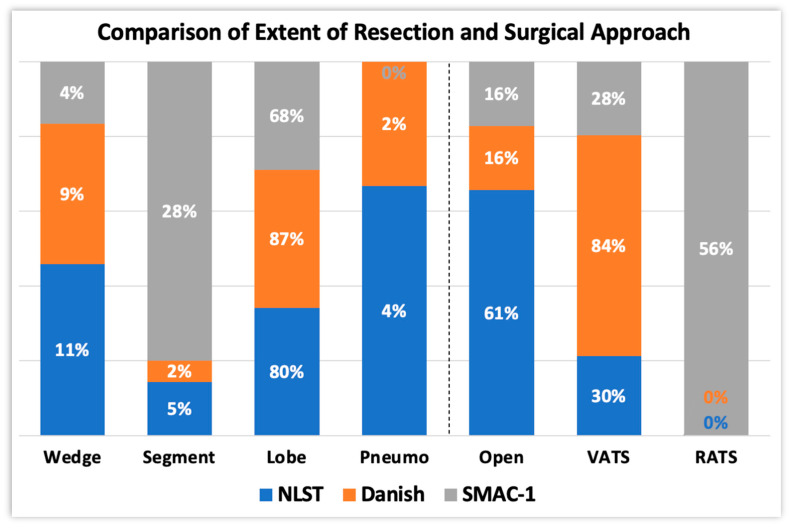
Comparison of extent of resection (**left**) and surgical approach (**right**) between SMAC-1 and two other lung cancer screening programs, carried out over 15 years ago.

**Figure 5 cancers-16-00417-f005:**
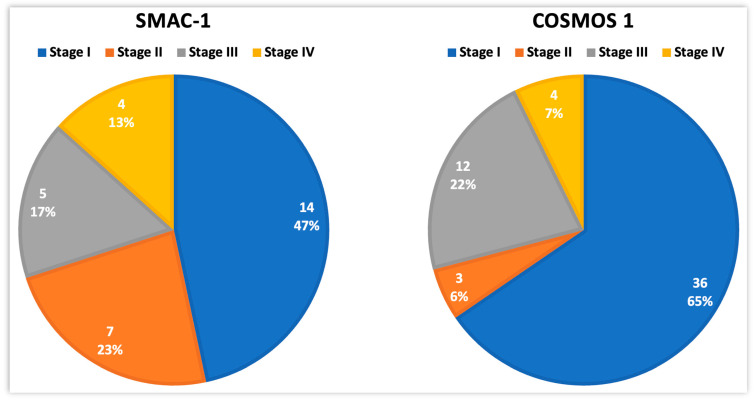
Distribution of disease stage at baseline in two consecutive screening studies performed in Milan, Italy (SMAC-1 and COSMOS 1). Total number of detected lung cancers was 30 and 55, respectively.

**Table 1 cancers-16-00417-t001:** Subjects’ characteristics. * Who had quit within 15 years (>30 pack/y); ** of these, two were extra-pulmonary tumors (one thymoma and one mesothelioma).

	N (%)	Mean ± SD	Median (IQR)
**Age (years)**		64 ± 6.6	64 (59.7–68.9)
**Gender**			
M	693 (62.3%)		
F	419 (37.7%)		
**BMI**		25.3 ± 4.1	25 (22.6–27.8)
**Smoking status**			
Active	862 (77.8%)		
M	508 (58.9%)		
F	354 (41.1%)		
Former *	250 (22.2%)		
**Smoking duration (years)**		42.5 ± 7.5	42 (40–48)
**Pack/y**		48.5 ± 19.8	45 (35.2–57)
**PLCO_m2012_ risk**		5.6 ± 4.9	4 (2.6–6.6)
**Personal history of cancer**	120 (11%)		
**Family history of lung cancer**	420 (38%)		
**Educational level**		2.47 ± 1.16	2 (2–4)
1	217 (20%)		
2	506 (46%)		
3	73 (6.6%)		
4	288 (26%)		
5–6	28 (2.5%)		
**FEV1 %**		91 ± 17.7	92 (80–103)
**Emphysema**			
Mild	209 (19%)		
Moderate	128 (11%)		
Severe	47 (4.2%)		
**Maisonneuve (baseline LDCT)**			
All	1112 (100%)	1.02 ± 2.5	0.47 (0.33–0.75)
Negative	762 (69%)	0.44 ± 0.22	0.39 (0.29–0.54)
Indeterminate or suspicious findings	350 (31%) **	2.3 ± 4.19	0.89 (0.56–1.97)
Maximum diameter (mm)		8.14 ± 7.43	5.86 (4.5–8)

**Table 2 cancers-16-00417-t002:** Morphological visual descriptor of LDCT-scan-detected lung nodules, according to their maximum diameter.

(a)			(b)	
Size (mm)	Solid 233 (69%)	Partially Solid 69 (20%)	Size (mm)	Non Solid 39 (11%)
<6	136	35	<6	11
6–8	53	19	6–14	6
>8	44	15	>14	22

**Table 3 cancers-16-00417-t003:** Surgical approach, extent of resection, histology, and pathological staging of the 30 patients diagnosed with lung cancer. * The 8th Edition of TNM was used.

		N (%)
**Surgical approach**	Open	4 (16%)
	VATS	7 (28%)
	RATS	14 (56%)
**Extent of resection**	Wedge	1 (4%)
	Segmentectomy	7 (28%)
	Lobectomy	17 (68%)
**Hystology**	Adenocarcinomas	23 (76.8%)
	Squamous cell carcinomas	3 (10%)
	NET Small cell	2 (6.6%)
	NET carcinoid	2 (6.6%)
**pStage ***	I	14 (47%)
	II	7 (23%)
	III	5 (17%)
	IV	4 (13%)

## Data Availability

De-identified data presented in this study are available on request from the corresponding author. The data are not publicly available due to privacy and ethical restrictions.

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
