# Peer review of "The Smokers Health Multiple ACtions (SMAC-1) Trial: Study Design and Results of the Baseline Round"

_cancers, 2024, doi:10.3390/cancers16020417_

Round 1

Reviewer 1 Report

Comments and Suggestions for Authors

NICE WORK.

ABSTRACT OK

INTRODUCTION OK.ADDING THE CLASSIFICATION OF WHO 2021 WOULD BE AN EXTRA POINT!

METHODS OK

RESULTS OK.JUST REFER THE PROBABLE PATHOLOGY OF THE  NON CANCER NODULES.

DISCUSSION  OK

CONCLUSIONS  OK.MAYBE A MORE CLEAR SUGGESTION/GUIDELINE PROPOSAL FOR THE USE OF SMAC-1(ONCOLOGY ROUTINE IN RELATION WITH ECONOMY)

Author Response

Thank you very much for your feedback.

1) Unfortunately, we do not have sufficient information regarding the WHO 2021 classification of lung tumours captured in the database used for this paper.

2) We report 0 false positives at surgery, thus there is no probable pathology of the 'non cancer nodules' that can be calculated.

3) The multi-component feature of SMAC-1 provides policymakers with an opportunity to implement screening at a national level. Nevertheless, rather major investments are required upfront to even get started at such, larger scale, level. A dedicated paper featuring economics aspects derived from our SMAC-1 experience is underway. Please, see dedicated text added at lines 451-455.

Reviewer 2 Report

Comments and Suggestions for Authors

Congratulations on completing a very important project. However, a few things need attention:

1) It is not clear to the reader from your paper if you have collected substantial and useful data about the cardiovascular risks, and what will be done about that. Regarding molecular markers, it is clear results will be discussed in forthcoming papers, but cardiology findings and further proposed actions are not described. If this will also be addressed in a future paper, should be clearly stated.

2) what other studies have looked at cardiac risk assessment with LDCT? a brief description and citations would be helpful. 

2) Why do you think engagement from GPs was so low in spite of measures you took, like sending e-mails and organizing conferences? Some insights and discussion on this point would be helpful.

Comments on the Quality of English Language

Minor revisions, but maybe just European vs North American differences

Author Response

Thank you so much for your input.

1) CV risk data have been collected and a dedicated paper is underway, as stated in the text, please see lines 291-293 (furthermore, an abstract was presented at the last IASLC meeting in Singapore, ref. 25).

2) Done, both in the text (lines 89-100) and in the citations (16-18 and 20-23 and 25).

3) It is because GPs (WE ALL, after all) work in the era of the paper-work crisis. Proper text and citations have been added. Please, see lines 331-334 and ref. 64 and 65.

4) Yes, NA english style indeed.

bis) To provide you with a better background on our work, a statistical short paragraph has been added (see lines 226-231). To better present the results, table 2 has been revised.

Reviewer 3 Report

Comments and Suggestions for Authors

The authors present a strong study, well described, that deals with the critical issue of early detection of lung cancer risk with convincing data.

there are several additional questions that might be of interest for them to comment on in the paper:

1. the location of the nodules, including clustering, etc that might be observed

2. when serial studies have been carried out, considering the classification concerning solid vs non-solid, have transitions between states been noted

3. in serial studies, how common, or not, was the appearance of additional nodules

4. when serial studies were performed, what were the percentages of nodules that underwent change

Author Response

Thank you very much for your comments!

1) Location by lobes has been added in the text (lines 248-251)

2-4) Lung nodules consistency has been captured at the first available radiological diagnosis. Radiological changes over time has not been recorded. Nevertheless, we understand and agree with your valid points. We will keep this into consideration for future studies and sincerely thank you for your input.